# Schooling of light reflecting fish

**Assaf Pertzelan** [1,2] *, **Gil Ariel** [3], **Moshe Kiflawi** [1,2]

**1** Faculty of Life Sciences, Ben Gurion University, Beer-Sheva, Israel, **2** The Interuniversity Institute for Marine Sciences at Eilat (IUI), Eilat, Israel, **3** Department of Mathematics, Bar-Ilan University, Ramat-Gan, Israel

\* assaf.pertzelan@gmail.com

## Abstract

One of the hallmarks of the collective movement of large schools of pelagic fish are waves of shimmering flashes that propagate across the school, usually following an attack by a predator. Such flashes arise when sunlight is reflected off the specular (mirror-like) skin that characterizes many pelagic fishes, where it is otherwise thought to offer a means for camouflage in open waters. While it has been suggested that these 'shimmering waves' are a visual manifestation of the synchronized escape response of the fish, the phenomenon has been regarded only as an artifact of esthetic curiosity. In this study we apply agent-based simulations and deep learning techniques to show that, in fact, shimmering waves contain information on the behavioral dynamics of the school. Our analyses are based on a model that combines basic rules of collective motion and the propagation of light beams in the ocean, as they hit and reflect off the moving fish. We use the resulting reflection patterns to infer the essential dynamics and inter-individual interactions which are necessary to generate shimmering waves. Moreover, we show that light flashes observed by the school members themselves may extend the range at which information can be communicated across the school. Assuming that fish pay heed to this information, for example by entering an apprehensive state of reduced response-time, our analysis suggests that it can speed up the propagation of information across the school. Further still, we use an artificial neural network to show that light flashes are, on their own, indicative of the state and dynamics of the school, and are sufficient to infer the direction of attack and the shape of the school with high accuracy.

## 1. Introduction

Free-ranging animals that move in large aggregates, such as schooling fish and flocking birds, are often required to maneuver in unison to evade attacking predators. The collective evasive maneuvers change macroscopic properties of the aggregate, such as its direction, polarization, and density [1–4]. These changes are often manifested as "waves of agitation" or "escape waves" that follow the propagation of information regarding the attack across the aggregate. The complex inter-individual interactions, which bring about such large-scale dynamical patterns, are not fully understood [2–4]. One of the main empirical obstacles to a fuller understanding of these interactions and the consequent escape dynamics is the difficulty of tracking individuals within very large schools; using either cameras (e.g. [5]) or sonars [2, 5].

**Data Availability Statement:** All relevant data are within the paper and its Supporting information files.

**Funding:** The author(s) received no specific funding for this work.

**Competing interests:** The authors have declared that no competing interests exist.

**Fig 1. Propagation of an agitation (shimmering) wave in a school of _A. lacunosus_ (top-right corner), filmed at 60 frames per second.** The flashes arise when the fish, initially facing to the left, turn into the school (here into the page, perpendicular to the advancing wave), causing their body to face the observer at just the right angle to generate a flash of reflected sunlight. The wave propagates as the decision to turn is taken sequentially by the fish, starting from the left. Bottom-left corner: a "predator's view" of the wave, taken from a camera mounted on a spear shot into the school. The video clips, from which the snapshot were taken, were filmed while snorkeling along the shoreline of the Golf of Eilat, using Sony RX100 IV and Paralenz Dive Cameras. See S1 and S2 Videos for the actual video clips.

Large schools of silvery fish are common in the world's oceans [6]. When swimming close to the surface, the specular (mirror-like) skins of these fish will often reflect direct sunlight. To an underwater observer, these reflections appear as highly conspicuous flashes of light (Fig 1), which increase the fish's contrast by at least one order of magnitude [7]. When under attack, an agitation wave crossing a school of specular fish, may appear as a shimmering wave of flashes (e.g. Fig 1). The wave arises as the succession of evasive maneuvers by neighboring school members momentarily brings their bodies to an angle that reflects the sun in the direction of the observer [1–4]. As such, shimmering waves could contain information regarding the dynamics of the school and individuals within it, which is more discernable than the actual trajectories of the individual fish. For example, the dark bands typical of agitation waves in starling flocks, which are analogous to shimmering waves, have been used to measure wave intensity and propagation speeds [8].

In this paper, we use individual-based simulations to study the dynamics of shimmering-waves. We begin by testing whether a shimmering wave can arise in possible schooling scenarios. We continue by addressing the possibility that flashes observed by the fish themselves can be used to induce apprehension and, thereby, speed-up the transfer of information within the school. We then continue to test the complementary question of whether the dynamic properties of the waves (e.g., their duration and speed of propagation) can be used to differentiate between competing hypotheses relating to the inter-individual interactions that give rise to them; as well as to infer the occurrence and direction of an attack.

## Fish schools under attack

In schooling fish, the response to an attack amounts to moving closer together or rolling sideways [1–3]. Once initiated, waves of agitation often propagate faster than the speed at which individuals are moving within the group; a phenomenon often referred to as the 'Trafalgar

effect' [9]. In cases where the stimulus is an approaching predator, the waves have also been found to travel faster than the predator itself [4]. Surprisingly, waves can travel even faster than expected given the estimated response-latencies [3], i.e., the time between the perception of the stimulus and the fear/escape response. The unexpected speed of information-transfer led to speculations regarding possible mechanisms that enhance synchrony and extend beyond the scale of localized social interactions. A prominent example is the "chorus-line hypothesis" [10], which assumes a reduction in latency due to a heightened state of anticipation.

Importantly, the flashes of light reflected off specular fish are visible not only from outside the school but also from within it; and thus, could potentially serve to inform the school members themselves. Indeed, it has been previously hypothesized that the reflective structures on fish can be used for "communicating information on relative positions, orientations, and movements between neighbors" [11]. Here, we explore a hypothesis that the heightened state of anticipation described above is caused by changes in the pattern of reflected light, as perceived by school members found downstream of the propagating wave. Particularly, we propose that observing a large change in the number of flashes reduces the latency and, as a result, speed-up the propagation of information.

## Modelling schools under attack

Standard models of schooling typically fall short of generating agitation waves following localized perturbations, such as an attacking predator; presumably because the perturbation is perceived by only a small number of agents [4, 12]. In most models (for example the three-zones model e.g. [13–15]), individuals modify their position and direction based on the average response of neighboring agents. As a result, the initial reaction to the perturbation is "averaged out" [1–3, 12], i.e., the response of an agent close to the perturbation is averaged with agents that are farther away and, thus, did not perceive it. As most agents are far from the perturbation, the intensity of the reaction decreases with distance. Indeed, subsequent models have shown that additional responses, that are not due to a direct observation of the predator but leads to preemptive evasive maneuvers, are needed in order to generate agitation waves [3, 8, 12]. For example, [12] assumed that when an agent sees another agent whose behavior is clearly different from other school members, it will copy its behavior. With this added behavioral component, information of an approaching predator can propagate through the school, forming a response wave that travels at a (approximately) constant speed [4]. The agitation wave gives rise to a collective evasive response, even though the number of individuals that directly experience the perturbation is small [12].

The simulations introduced below accommodate the three basic local-interaction rules which are typically included in collective-motion models (repulsion, attraction, and alignment), as well as the copy response suggested in [12]. On this collective-motion model we superimpose a ray-tracing model [7] that 'records' the light-flashes perceived from a pre-prescribed location, either within or outside the school. To the best of our knowledge, our model is the first attempt to model shimmering waves, based on first principles of light propagation and reflectance.

The paper is organized as follows. Section 2 describes the basic methodology underlying our modeling and simulation. More specific details are provided at the beginning of each of the three subsequent sections (3,4 and 5), and in the S1 File. Sections 3,4 and 5 focus on the results pertaining to our three main objectives. In section 3, we demonstrate that the model successfully produces shimmering waves using realistic parameter values (Table 1); contingent on the inclusion of a copy response, i.e. consistent with [12]. We then test, in section 4, the hypothesis that the shimmering wave caused by an attack can: 1) be seen by fish found

**Table 1. Translation of the simulation parameters to real-world values.**

| Parameter name | Parameter value | Source |
|---|---|---|
| Wave speed [m/sec] | 10 | [1, 17] |
| Wave Distance Per Frame [m] | 28/100 | Internal simulation parameter |
| Number Of Frames Per Second | 35 | Internal simulation parameter |
| dt (iteration time) [sec] | 0.028 | Internal simulation parameter |
| Latency [sec] | 0.084 | See [18–21] |
| Latency After Seeing Signal [sec] | 0.028 | The iteration time dt |
| Fish Routine Speed [mm/sec] | 107 | 1.07 BL |
| Fish Escape Speed [mm/sec] | 1070 | 10.71 BL. Within the range reported by [21] |
| Fish Body Length [mm] | 100 | assumed for translating our simulations to realistic values |
| Rotation Rate Lower Bound [degree/msec] | 6.428 | [21] |

downstream from the propagating wave; and, were the fish to use this information, 2) leave a discernable signature in the dynamics of the waves. In section 5 we demonstrate the ability to distinguish between different characteristics of the school, based only on observed flash patterns. The characteristics we focus on include: the direction of the attack, the shape of the school, and the underling rules of motion used by the individual; and in particular the use of the flash signature as input. We discuss our results in section 6.

## 2. General methods

Our modeling strategy is a second-order agent-based model (i.e., we specify the acceleration). Simulations include a school of fish and a single predator. The latter appears for a limited number of steps and moves along a fixed linear trajectory. The predator moves towards and into the school faster than the agents' maximal speed during the 'schooling' state (see below).

In the three-zones model (e.g [13, 14]), each agent $i$ at every discrete time-step $t$ is described by its position, $p_i(t)$, and velocity, $v_i(t)$. To account for predator response (the *direct-only* model), we introduce two additional variables: the acceleration $a_i(t)$, and a discrete internal state $s_i(t)$. The internal state is one of four options: schooling, evasive, predator-response (similar to [16]), or copy-response (similar to [12]), as described below. Moreover, in order to trace the direction at which light-rays reflect off the fish, we keep track of the vertical direction of each agent. To this end, we define a direction $b_i(t)$ that is perpendicular to the velocity, corresponding to the direction of the dorsal side of the fish (the direction that regularly points upward, perpendicular to the vertical axis).

Below, we outline the main components of the model. Details are provided in the S1 File, section 1.

### Agent dynamics

At each step, all agents update their positions and velocities synchronously as,

$$p_i(t + 1) = p_i(t) + v_i(t)$$

$$v_i(t + 1) = Iv_i(t) + a_i(t),$$

where $I < 1$ is an inertia parameter, qualitatively accounting for water resistance. Velocity-

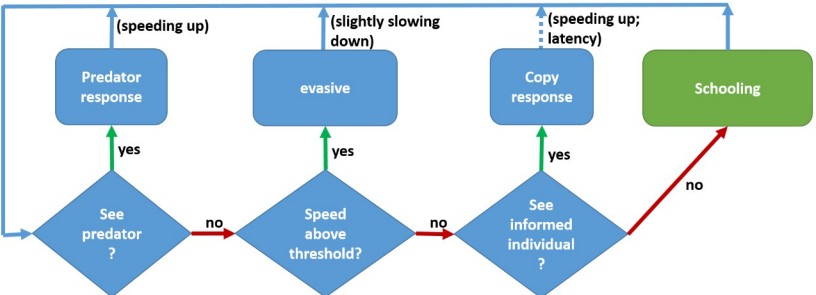

**Fig 2. State diagram.** A diagram of the decision process for moving between states, $s_i(t)$. The states are represented by rectangles and the conditions by diamonds. If an agent sees a predator, it accelerates to a high speed and turns away from the predator. If an agent sees an informed individual (a high-speed individual facing away of the agent direction), it copies the velocity of this individual. From then on, it remains in the *evasive* state until it gradually slows down and return to the *schooling* state. The copy response occurs after several steps of latency (default: 3 steps; corresponding to 0.084 msec. See Table 1), during which the agent remains in the *schooling* state.

updates depend on the current internal state $s_i(t)$, according to the decision process explained below and depicted in Fig 2.

See Table 1 for conversion of the simulation parameters to empirical estimates.

## Internal states

- *Schooling*: An agent's acceleration is determined similar to the usual three-zone rules, e.g. [22], i.e. a weighted average of the attraction to all neighbors within a given distance $R_o$, alignment with all neighbors within a distance $R_a$ and repulsion from neighbors within a distance $R_r$; with $R_r < R_a < R_o$ (S1 Fig).

- *Evasive*: If an agent in the *schooling* state attains a speed that is higher than the schooling speed limit $|v_i(t)| > s_{max}^{schooling}$ (for example, because it encountered a predator or an informed individual–see *copy-response* below), it transitions to an *evasive* state. In this state, the update rules for the velocity similar to rules while *schooling*, but with different weights (S1 Table in S1 File) that give a higher priority to cohesion and alignment over repulsion.

- *predator-response*: If agent in the *schooling* or *evasive* states encounters the predator (i.e., the predator is within a distance $R_p$), it transitions into the *predator-response* state. In this state, agents turn towards the opposite direction of the predator and start swimming at speed $s_{max}^{evasive}$.

- *copy-response*: A neighbor (to any *schooling* agent within radius $R_i$) is called *informed* if it is currently making a sharp turn and is swimming fast (above a threshold). Such a behavior indicates (possible) information of an attack. A *schooling* agent that has an informed neighbor will transition into the *copy-response* state following a response delay (latency). In this state, agents copy the velocity of the informed neighbor. In the following, we set $R_i = R_a$, as both represent the distance at which the orientation of neighbors are visible.

## Roll angles

To track the angle at which incident light beams are reflected off the agents, we need to model the dynamics of a direction normal to the velocity, corresponding to the ventral side of the fish. We denote this direction as $b_i(t)$. To this end, we define a motion rule for the roll-

alignment, which determines the torsion of the trajectory curve of each agent. In addition, we give individuals a tendency to realign $b_i(t)$ with the vertical axis (point upwards). See sections 1 in the S1 File for details.

## Optics

We assume a light source above the water-surface that is a cone of light of angle $\theta$ around an incident direction $d_{\text{light}}$. In all simulations, we take $d_{\text{light}} = -i_3 = (0,0,-1)$, i.e., pointing directly downwards. For simplicity, the agents are considered as 2-sided planar mirrors whose normal direction $n_i(t)$ is perpendicular to the velocity and back vectors,

$$n_i(t) = \frac{v_i(t) \times b_i(t)}{|v_i(t) \times b_i(t)|}.$$

A more detailed model that provides a realistic representation of light reflected off the skins of silvery fish is given in [7]. This model, which is computationally expensive, was only used for static snapshots of fish schools with no dynamics. In order to determine if agent $i$ reflects light towards position $x$ in a given time $t$, we define a vector in the direction of $p_i(t) - d_{\text{light}}$,

$$ray_i(t) = \frac{p_i(t) - d_{\text{light}}}{|p_i(t) - d_{\text{light}}|}.$$

Then, we define the reflected vector of $ray_i(t)$ from the plane represented by the normal $n_i(t)$,

$$ref_i(t) = ray_i(t) - 2[ray_i(t) \cdot n_i(t)]n_i(t),$$

and another vector pointing from the agent position towards a virtual observer that is placed at position $x$,

$$obs_i^x(t) = \frac{x - p_i(t)}{|x - p_i(t)|}$$

If the angle between $ref_i(t)$ and the vector $obs_i^x(t)$ is smaller than $\theta$, then the agent reflects light towards $x$. It is possible that an agent will not reflect light both at time $t$ and at time $t + 1$, but its trajectory for its location between the two times will cross a point in which it will flash. This is taken into account, as explained in the S1 File and in Fig 3).

## Classification of behavioral patterns

To test if shimmering waves can be used to differentiate between some of the behavioral and structural configurations accommodated by the model, we apply an artificial deep neural network (DNN) to the 'recorded' pattern of reflections. DNNs are increasingly used in the study of collective animal behavior [23–27], owing to their capacity to find features that are intuitive to recognize but unintuitive to define. In particular, DNNs are efficient in detecting features in images taken from different perspectives and scales [28] and in sequences of inputs [29–31]. Since the training of DNNs requires large, annotated datasets, researchers are increasingly turning to synthetic data [24, 26, 27, 32, 33]. We adopted a similar approach, using our model to facilitate supervised leaning by the neural network.

## 3. Generation of flash waves

Our modeling approach enables us to demonstrate that a model implementing both *predator-response* and *copy-response* can successfully generate seemingly realistic shimmering waves.

**Fig 3. Light reflection.** The angular distribution of the incident light rays above the agent is represented by the light blue cone. The agent, represented by the gray elliptical cross-section of a forward facing fish in the middle of the frame, has its side facing an observer (e.g. predator) situated to its left. The solid lines represent the incident and the reflected ray reaching the observer. a. A "flashing" agent. The reflected ray enters from within the light cone. b. The reflected ray is outside of the light cone, so the agent does not appear as flashing. c. A "continuous reflection" event. The dashed ellipse in the middle of the frame represents a non-flashing agent at time $t$-1 and the gray ellipse line represents a non-flashing agent at time $t$. If the angular trajectory of the agent (represented by the gray arched arrow) reflects a ray that passes through the cone of incident light, it would flash toward the observer sometime between $t$-1 and $t$, and would be counted (by the simulation) as a flash at time $t$.

Moreover, the copy response is necessary, i.e., the basic three-zone interaction rules and *predator-response* alone are not sufficient to explain the waves seen in nature.

## Methods

All schools are composed of $N$ = 15,000 agents, initially positioned in an elongated cylinder parallel to the x-axis (assuming a body length of 10cm and a density of ~33 fish/m³, an average distance of ~40cm between agents, corresponding to a cylinder of around 28m long and 5.5m wide; see S1 Table in S1 File for the full configuration).

A virtual observer was placed ~12.5m from the school center of mass on the same horizontal plane and perpendicular to the school average velocity, i.e., the observer is watching the school from its side (Fig 4). During each scenario, the observer remains at the same relative position to the center of the school (that is practically stationary) but not moving relative to the orientation of the school. The predator attacks the school head-on and aims its attack at the center of mass of the first 30 individuals, evaluated in relation to the average velocity of the school. The predator was removed from the system at the end of the attack, which lasts five time-steps. Simulations resolve the group dynamics for 115 simulation steps, corresponding to approximately 4 seconds (see Table 1).

We compared four response models that differ in their velocity-update mechanisms:

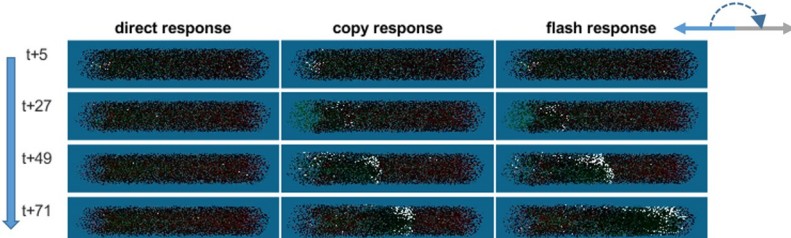

**Fig 4. Snapshots of the dynamics that develop following a head-on attack on a cylindrical school swimming to the left, under each of the three models (columns).** Time-steps are shown on the left, and progress from top to bottom. The shimmering wave appears as fish reflect light (turn 'white') toward the observer, as they turn approximately 180 degrees (arrows at top-right corner) to face the other direction and evade the predator.

- *Model 1*: *Direct-only*: A fish in the schooling state that encounters the predator directly performs an evasive response that increases their speed (see Section 2). Under this model, the fish react only to a direct observation of the predator.

- *Model 2*: *Direct-and-copy*: In addition to the direct response, fish found in the schooling state that do not see the predator, but do see an informed individual, enter a *copy-response* state and copies the velocity of the informed agent. Copying occurs with a latency of 3 steps (Table 1), corresponding to the reaction time of a fish.

To quantify the dynamics that follow the attack, under each of the four models, we first divided the school into ~300 bins (slices) along the x-axis, each corresponding distance of 100mm. For each bin, at each time-step, we then calculated three measures:

1. Change in local fish density: The number of fish in bin *i* at time *t*, expressed in standard deviations (z-score, using the mean and variance calculated across time-steps). As in previous works [3, 4, 34], we expected a wave of temporarily increased numbers (i.e., density) to propagate across the school.

2. The proportion of fish that were perceived as flashing by the external observer. This is relevant for establishing the emergence of shimmering waves as the result of the escape maneuvers.

3. The proportion of fish that encounter at least one informed individual, at time *t*. This measure quantifies the actual propagation of information.

Each of these measures was plotted, using a heatmap as a function of both the position along the *x*-axis of the school and time. Patterns in these heatmaps were used for inferring the dynamics of the respective measure. In particular, the presence of a diagonal in a heatmap was taken as indicative of the propagation of the respective measure, with the slope of the diagonal equaling the reciprocal of the speed of propagation.

## Results

As expected, a localized predator attack under the *direct-only* response model did not produce a shimmering wave (Fig 4, left column), indicating a rapid loss of information due to averaging-out of the evasive response. Indeed, all measurements under the *direct-only* model do not reveal any meaningful patterns or indicate propagation of information (See S1 File section 8 for the plots).

In contrast, the same attack under the *direct-and-copy* response model produced the expected shimmering wave, which traversed the school at an approximately constant speed (Fig 4, middle column). Neighbors of informed individuals copy its behavior, without loss of the information regarding the direction of the attack. As a result, the copied response propagates through the entire school within 115 steps, corresponding to 3.22 seconds (see Table 1).

The *direct-and-copy* model produced a density wave, which appears as a diagonal in Fig 5a. The propagation of the density wave is matched by the pattern of flashes perceived by the external observer (Fig 5b), which follows the actual propagation of information across the school (Fig 5c). Thus, the rate at which fish were exposed to an informed individual (and copied its behavior) matches the density and shimmering waves.

The supplementary material (S1 File section 10) details a sensitivity analysis which shows that wave speed decreases with latency and increases with escape speed (the initial evasive speed. Moreover, (S1 File sections 11, 12) the shimmering wave is not sensitive to the exact location of the observer, the shape of the school and, to some degree, the alignment of the school.

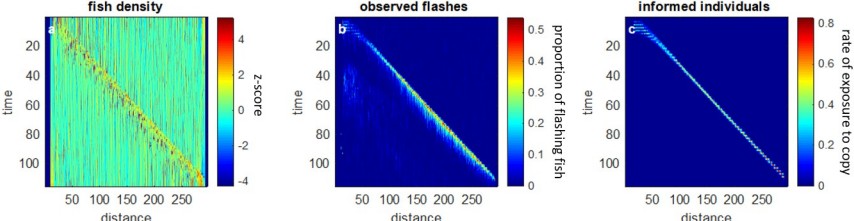

**Fig 5. The *direct-and-copy* model, with latency of 3 time-steps.** For each plot, the x-axis represents a spatial division of 3D-space along the x-axis, which corresponds to the length of the school. The y-axis represents the timeline, starting with the attack at t = 0 and running for a total of 115 steps. a A wave of changes in density(z-score) progresses with a constant speed along the school. The leftmost part ($x$, $t = 0$) is distorted due to direct predator responses. b. The proportion of flashing fish in each bin, as seen by the external observer. The flash pattern which is visible to the observer is consistent with the density wave. c. The actual propagation of information, i.e. the rate of new fish that were exposed to an informed individual and copied its behavior. The slope of the diagonal represents the speed of the wave.

## 4. Possible use of the flash patterns by the group members

### Methods

In addition to the appearance of the flash waves to an external observer, we wanted to test whether shimmering patterns could be used as a source of information by the group members themselves. Therefore, we introduced a new assumption/hypothesis–namely, that fish situated within the school can perceive and act upon changes in the number of flashes. To incorporate such a 'flash response', we assume that intense shimmering–i.e., the turning 'on' or 'off' of a large number of flashes, per time unit–can signal an instability within the school. We will refer to such an occurrence as a *flash-signal* (see S1 File for the precise definition). We further testing the assumption that agents respond to this signal and, as a precaution to a possible attack, enter an anticipation mode that shortens their response-latency which, thereby, results in a faster evasive maneuver in response to local interactions [35]. We modified the *direct-and-copy* model by specifying two variations of how a fish respond to an abrupt change (increase or decrease) in the number of flashes, which falls above some specified threshold (+/- 200):

- *Model 3*: *flash-latency*: Following a *flash-signal*, the fish reduces the latency of its response to zero.

- *Model 4*: *flash-direct*: Following a *flash-signal*, the fish turns away from the center of mass of the signal.

### Results

With the *flash-latency* model (Fig 4, right column), the initial dynamics are similar to those seen with the *direct-and-copy* response model. However, at some point ($\sim t = 40$) the fish downstream of the propagating wave perceive an above-threshold change in the overall flashing pattern which shortens their latency and consequently, speed up the propagation of the wave (Fig 6).

Given the high detectability of the flashes, the number of flash changes is visible in all areas of the school, including those not yet reached by the propagating wave of informed individuals. As a result, the wave can be used as an early warning. Fig 7 demonstrates how the flash signal is viewed by a virtual fish at the back of the school.

With the *flash-direct* model, an abrupt simultaneous response of all individuals to the predator occurred which resulted in a quick flash cloud. See S1 File and S12 and S13 Figs write column for further information.

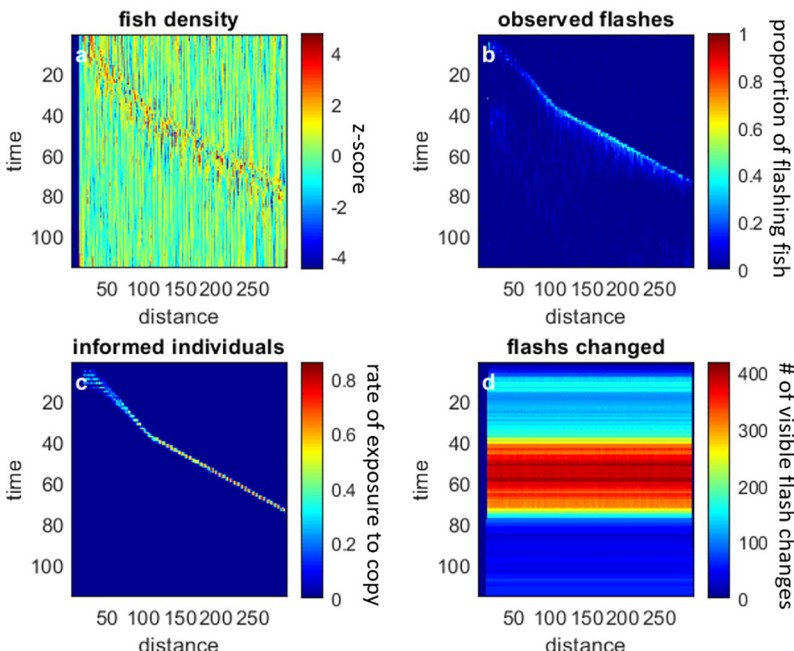

**Fig 6. The four basic measures for the *flash-latency* model.** The wave starts at the same speed as the copy-response scenario (compare plots a, b, and c to their corresponding plots in Fig 5) but changes around $t = 40$. At this point in time, the flash signal crosses a threshold (set to 200; plot d) for all fish, which reduces the response latency, accelerating the propagation of information.

## 5. The information content of shimmering waves

The previous section describes several agent-based models that recreate shimmering waves as a response to predator attacks. The goal was to generate flash patterns that resemble natural waves and explore how fish may respond to them. Here, we take a complementary approach and demonstrate that flash patterns themselves contain enough information to enable an

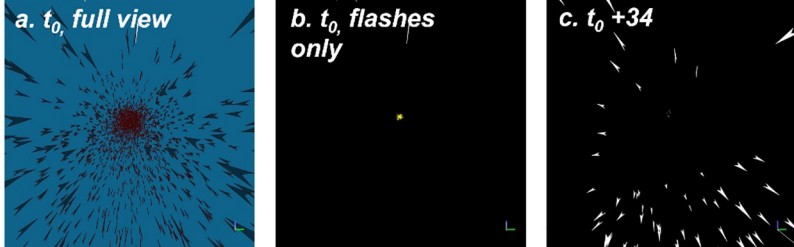

**Fig 7. The view from the back of the school ($N = 5000$, inter distance between fish 400mm).** Plot a. shows the full-detailed view of the school, one time step before the attack. This view, which shows the fish rather than the flashes, cannot be seen fully by the fish due to the low contrast of the non-flashing school members. Plot b is the flash-only version of plot a, showing the predator in the distance (in the middle of the frame) but no visible flashes of yet. After 34 steps (plot c), the flash wave gets close to the observing fish. For illustrative purposes we chose to use relatively sparse schools, placed the observing fish at the "back" of the school and present a narrower field of view than that which is available to the simulated observer. For clarity, we ignore the occlusion of flashes; as we do also in the models. In section 7 of the S1 File we discuss the consequences of excluding occlusion in this scenario (see also the insert at the bottom-left of Fig 1, for a real-world example of the flashes seen from within the school).

observer to differentiate between scenarios; i.e., the observer could infer differentiate between scenarios that pertain to the behavior of the fish within the schools, or the structure of the school as a whole; without seeing the individual fish themselves. In other words, while in the previous sections we explored properties of flash-patterns that could be used, in this section we aim to provide an example in which such differentiation is indeed possible. As proof-of-concept, we focused on three analytical tasks: detect an attack and its direction, infer the shape of the school under attack and infer the motion-rules of the individual fish (Fig 8).

## Methods

The main idea is to train the observer to classify global states of the school into several categories, as listed below. To this end, we train a neural-network (NN) which only has as input the sequence of light flashes that a stationary observer from outside the school would perceive. We defined three tasks. In each task, the NN needs to distinguish between several classes, as detailed below.

*The occurrence and direction of attack* (Task 1). We consider six classes, corresponding to attacks from five different directions (relatively to the fish head): front, back, side, top, and bottom, and an additional "no-attack" scenario (Fig 8a). The NN needs to determine whether an attack occurred and if so, from what direction.

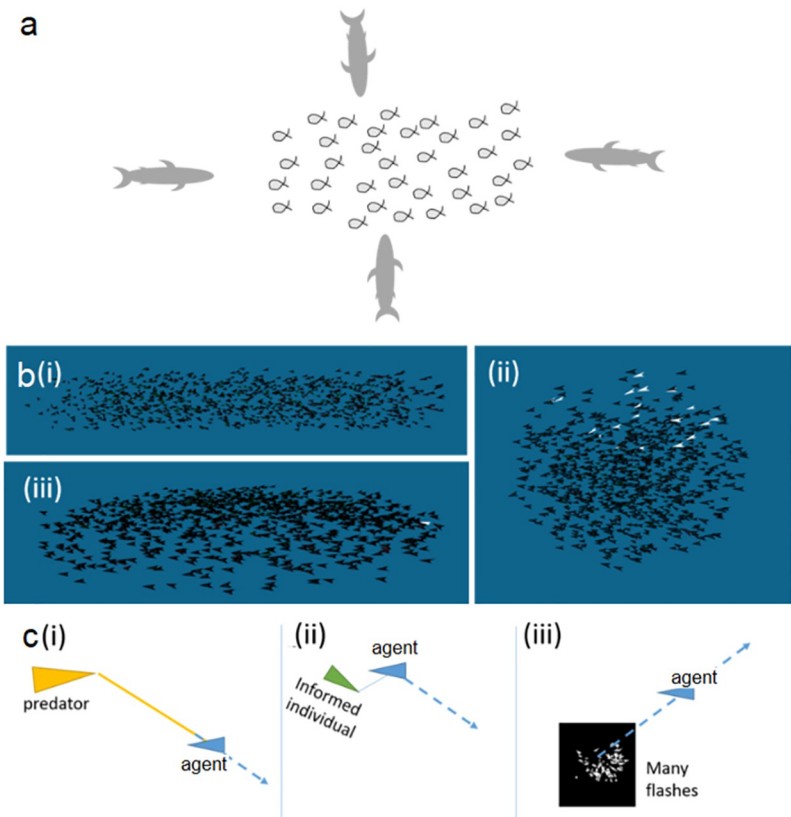

**Fig 8. Classification with respect to three parameters.** a. The direction of attack: front, back, top, bottom, side, and no-attack. b. The shape of the school prior to the attack: cylindrical, ball-, and pancake-shaped ($N$ = 1000 fish) c. The response to the attack: (i) direct-only (ii) direct-and-copy (iii) flash-direct.

*The shape of the school* (Task 2): We consider attacks on schools of three different shapes: elongated cylinder, ball, and pancake (a thin circular school close to the water surface). The NN needs to determine (using only the flashes) what the shape of the school was.

*The motion rules used by the fish* (Task 3): The scenarios considered in this task are the original response models (direct-only and direct-and-copy) and the two modifications; flash-direct and flash-latency. In other words, this task tests whether the use of light flashes as a precautionary signal by the fish will leave a discernable signal in the shimmering wave generated by the collective evasive maneuver. The NN, which in this task plays the role of an experimentalist studying the behavioral rules of schooling fish, needs to determine which of the models were used to generate the flashes.

In all simulations, school-size was set to $N = 1000$ fish, all initially oriented in the same direction (Fig 8). Working with a school-size that is considerably smaller than the 15,000 used in sections 3 and 4 allowed us to increase the dataset size (the number of training and testing runs, which are our samples, pre class). Our underlying assumption is that emergent patterns that appear with small $N$ would also appear with larger (up to a point) $N$.

## Dataset generation

For each of the tasks we generated 150 samples per class using the *direct-and-copy* response model described above. Each simulated sample consisted of a short image-sequence (a 'video') of 15 frames, 90x90 pixels each (equivalent to 0.2–0.8 seconds). See, for example, S4 Fig, showing only the flashes that are visible to the observer.

In order to differentiate between the classes, we used a convolutional-LSTM network (Based on [29]). Intuitively speaking, convolutional networks are efficient for detecting variations of image features regardless of scale and position of the camera. LSTMs are useful for processing sequences by adding a time-dimension to the process and tying together features of frames in different times, based either on timespan or on a set of features that were detected in between. In our case, when the observer neither knows its exact position nor his movement in relation to the group, and when there is a strong temporal factor, the combination of convolutional and LSTM architectures is appropriate [29–31]. See 'Classification model' in S1 File for more details.

For each task, the trained network was tested on a new dataset that consisted of multiple samples of the different classes (around 30 samples per class). The network was requested to identify which instance belongs to which class. The results are presented using a confusion matrix, where network classifications are compared to the true classes. Where relevant, we also present the total accuracy of the task results and the f1-score. Briefly, accuracy is the rate of true detections (negative and positive) from the total number of samples. *f1-score* is a measure that accounts for two other measures–– precision and recall., where *precision* is the rate of true positives from the total positives detected and *recall* is the rate of true positives from the total instances. See S1 File section 5 for the full results and the definition of the indices.

## Results

Below we demonstrate our ability to infer the state of the school under each of the three tasks outlined above. The results were derived from analyzing the 'noisy' datasets, generated by models that included 'observer-orientation', 'look-at-moves', and 'roll-noise'. See the S1 File section 3 for results without noise.

**Task 1: Attack-direction.** The trained network can detect with high accuracy whether an attack occurred or not (f1-score of 0.92 for the no-attack scenario). It can also detect the attack direction, but with lower accuracy. With all noise parameters on, the overall accuracy of this

model is 0.74, which is better than chance (1/6) but indicates that there are still some difficulties in distinguishing between the classes (See S1 File section 5.3). These difficulties could be due to either essential similarities between the flash signatures, or non-optimal selection of hyperparameters (Fig 9a).

**Task 2: School shape.** The network detected the shape of the school with a high overall accuracy of 0.94 (See S1 File section 5.3). It should be noted that this level of accuracy is relevant to the school-size and observer-distance specified in the model, and may change if these parameters are changed (Fig 9b).

**Task 3: Motion rules.** The accuracy of this classification was 0.74, which is considerably larger than the 0.25 expected by chance. The identification of *direct-and-copy* and the *flash-latency* scenarios is good, and only slightly mixing with each other (f1-score of 0.58 and 0.68 respectively). Surprisingly, the *direct-only*, where no wave is generated, was mistakenly identified as *direct-and-copy* or *flash-latency* in five of the 28 samples (Fig 9c).

## 6. Discussion

The main goal of this work is three-fold. First, to establish whether models of schooling fish can generate realistic flash waves that propagate across the school in response to an attack. Second, to explore the possibility that school members are using this source of information themselves, and test how that can, in principle, affect the attack-response behavior. Third, to demonstrate that flash patterns indeed contain accessible information relating to the dynamics of the school, the behavior of individuals within it (in particular, their response to threats) and to the nature of the attack. Overall, the simulations provide a proof-of-concept which demonstrates that, on the one hand, flash patterns are indeed indicative of the state and dynamics of the school and the behavior of the individuals that compose it; and, on the other hand, that the flashes may be an important causative factor in shaping the escape behavior of the fish.

Models of collective motion typically view the propagation of information as proceeding at a highly localized scale (e.g. [36–39]). This raises fundamental questions on the efficiency of information transfer within schools. First, noise and measurement errors may add up over long distances so that the signal (or information), for example regarding an approaching predator, is lost or averaged out. For this reason, it has been hypothesized that natural flocks are in a critical state, which implies long range correlations, that can extend over the entire flock [40–42]. Such models are motivated by models of statistical physics in which the interaction is instantaneous [43]. Such an assumption is not realistic in natural groups, including fish

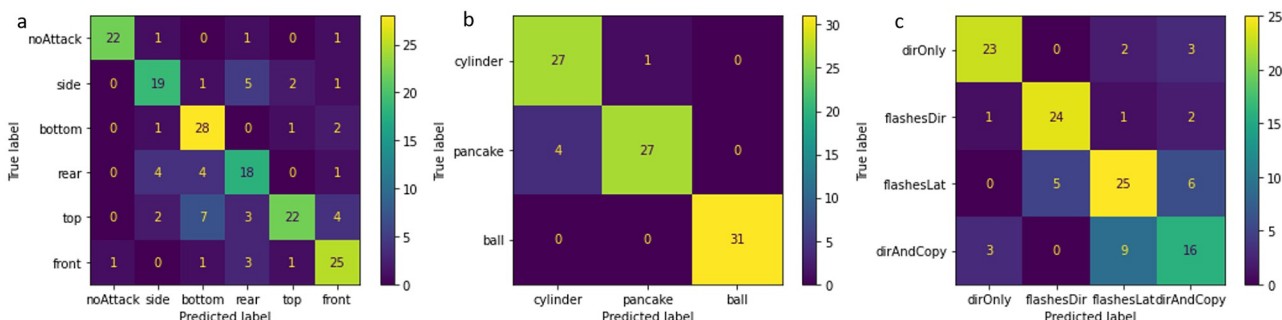

**Fig 9. Confusion matrices.** a. Direction of attack. The DNN could distinguish among different attack direction in most cases. b. 'school shape'. The DNN could distinguish among different school shapes in most cases. Some pancake-shaped schools have been mistaken for cylinders, possibly as both present a large school-depth when observed from certain angles. c. 'fish response'. The DNN could distinguish between different fish responses. The separation between the direct-and-copy and the flash-latency models is weaker than the ability to distinguish among other classes.

schools. For example, the reaction time of a typical fish is around 5–150 msecs [21]. This sets theoretical bounds on the possible speed of propagation of information within a school.

It has long been suggested that, at least in some examples, the actual speed of propagation may be higher [4]. To this end, based on observations of dunlin (*Calidris alpina*), Potts [10] suggested a "chorus-line" hypothesis according to which "individuals observe the approaching maneuver wave and time their own execution to coincide with its arrival". From a mechanistic perspective, this suggests long-range instantaneous transfer information, even if very limited. However, the rapid attenuation of light underwater (as opposed to the situation in air [44, 45]) implies that the mechanism suggested for birds, is not relevant for fish.

Our results suggest light flashes as a plausible mechanism for a "chorus-line" effect in near-surface schooling fish. As demonstrated in our simulations, light flashes, due to their high contrast [7], can act as macroscopic, long-distance, many-to-one emergent signaling. Moreover, it may play a role in synchronizing the schooling fish. We explore two hypothesis regarding the information fish can retrieve from a propagating wave of flashes, and their possible response to it. First, if the fish can infer the direction of an attack, as with the *flash-direct* model, fish can act accordingly and start preemptive measure to escape the attack. Such a response would alleviate the assumption of criticality because the information travels directly from its source, rather than from each individual to its neighbor. Alternatively, as with the *flash-latency* model, the critical state may still be required for the information to propagate across the school. In this case, fish only use the flash to increase the propagation speed, as discussed above. A similar idea was been proposed for flocks of starlings (*Sturnus vulgaris*) [46], in which the group converges to a marginal opacity state where the rate of visible sky provides to the individual information which is complementary to the one it gets from local interactions.

Our results show that flash-signatures convey enough information to differentiate between schooling scenarios. Event-detection, individual motion rules, properties of school geometries, and their combinations, could all be explored using the flash signature of schools. These results pave the way to empirical studies of flash waves and other flash-patterns. As we demonstrated using simulations, analyzing flash patterns as they appear for an external observer may also provide insight to the underlying decision-making process of the individuals, and hence bypass, at least partially, the need to track the trajectories of individuals. The abilities demonstrated in this study together with the fact that these flashes may be perceivable by the fish, lead to a possibility that a complex and rich flash-language is awaiting to be discovered, which is possibly already in use by the schooling fish and their predators.

## Supporting information

**S1 Data. The simulation and analyses codes.**
(ZIP)

**S1 Fig. The relative dimensions of the copy-response zone and the three social zones that give rise to schooling state.** The agent (the black fish in the middle) modifies its motion based on its neighbors' velocities and locations within the Cohesion zone (the agent is attracted towards the center of the mass of the neighbors' position within this zone, which is typically the largest). The Alignment zone (the agent adjusts its direction towards the average direction of its neighbors within this zone) and the Repulsion zone (the agent tries to get away of the agents in this zone by swimming away of their center of mass weighted by their relative distance from the agents' position. This zone, out of all zones, is typically the smallest and of the highest weight). Fish within the copy zone look for informed neighbors, whose swimming speed and direction they copy. Our default for this zone is the same as the alignment, assuming

that from this distance the fish can clearly see changes in orientation.
(TIF)

**S2 Fig. Example for the trajectory of a single agent, correcting its depth.**
(TIF)

**S3 Fig.** Wave speed as a function of parameters: a. latency, b. escape speed, c. response distances of all the motion rules (see default values in S1 Table in S1 File), d. the distance at which the fish applies the copy-response.
(TIF)

**S4 Fig. A comparison between the sequences of flashes generated by a simulated attacks on a cylindrical (top) and a spherical (bottom) school.** The attack came from the opposite direction to the school movement. The observer does not 'know' its position relative to the school (based on the flashes only) on the horizontal plane, and the center of the field of view of the observer is not fixed. Some of the flashes are due to random movement of the fish around their roll axis, while other are due to evasive or copy responses.
(TIF)

**S5 Fig. Uuncertainty parameters.** a. noise in the motion of the fish around the roll axis. b. noise in the look-at point of the observer. c. noise in the location of the observer on the horizontal plane around the school.
(TIF)

**S6 Fig. The classification_report output structure.** The names of the classes are presented in the top row by the order they appear in the table. For each class we measure the precision, recall, and f1-score. The total accuracy is being calculated for the entire model. Additionally, the *support* column presents the number of samples we used for testing each class. The last two rows present the weighted, and the non-weighted averages for precision, recall and f-score for the entire model. Since our dataset is balanced, the averaged f1-score is similar to the total accuracy, and there is not much difference between the weighted and the non-weighed averages.
(TIF)

**S7 Fig. Confusion matrix for attack direction, without uncertainty parameters.**
(TIF)

**S8 Fig. Confusion matrix for school shape, without uncertainty parameters.**
(TIF)

**S9 Fig. Confusion matrix for fish response, without uncertainty parameters.**
(TIF)

**S10 Fig. The effect of flash change with and without occlusions from a representative point at the back of the school.** a. The flashes the observer sees on time t+20 with occlusions. b. The flashes the observer sees without occlusions. c. The "real" image the observer sees. Note that the red and the green are visual aids for us indicating whether the fish turning towards the observer or away of it. They are not "seen" by the observer. Also, the image here is not limited by the distance the observer "sees" when it calculates its Boids rules. d. Correlation between the changes in total area of the flashes with and without occlusions in the different steps. Each plot (and dataset) was calculated on a different run but is replicable. It is also important to note that in this simulation the fish are two-dimensional and in real three dimensional schools the effect of occlusion may be stronger and depend in the school density.
(TIF)

**S11 Fig. The four measurements for the *direct-only* model.** The only pattern that could be clearly detected is the local predation event on the left side of the school at the beginning of the scenario. On plot d we can see that this event generated a weak flash signal towards the entire school.
(TIF)

**S12 Fig. *Flash-direct*.** Simultaneous response of the fish to the attack of the predator. The attack is followed by an instantaneous observed flash cloud and no information transfer afterwards. Since all the fish are changing simultaneously into an emergency state, there are no detections of informed individuals.
(TIF)

**S13 Fig. Comparison of low and high thresholds flash-response models to schools with no flash-response and "normal" (2) or fast (0) latencies.** A very sensitive school behaves practically as a school with low latency, while a non-sensitive school never activates its response to the flashes and acts like a normal school with latency 2. The right column shows the direct response to flashes which results in a fast explosion all over the school.
(TIF)

**S14 Fig. The 4 measures for short latency, as the one the fish are switching to when getting into emergency mode.** The dynamics are equivalent to those of the 'normal' latency but quicker.
(TIF)

**S15 Fig. When the school is very sensitive to changes in flashes, the latency is practically always 0.**
(TIF)

**S16 Fig. Low sensitivity leaves the school as if it would practically not respond to flashes.**
(TIF)

**S17 Fig. Plots showing the visible flashes for 5 different attack scenarios on 5,000 fish.** The escape response of the fish is to copy informed individuals and ignore flash signals. Plot a. Attack from the front of fish with full field of view. Plot b. Attack from the back of fish with full field of view. Plots c, d: Attack from front/back on fish with 85% field of view. Plot e. Attack from the front on fish with full viewing field while the escape response of the fish does not include turning into the group. In all plots we see that the attack leads to a flash wave in a constant speed (as in Fig 2 in the MS). The attacks from the back lead to a slightly weaker and less ordered wave but of the same speed as in the default configuration.
(TIF)

**S18 Fig. A flash wave as recorded from different angles horizontally around a cylindrical school.** The wave pattern is clearly seen except from perpendicular angles to the wave distance (plots 90 and 270). The closer the angle to perpendicular, the less linear the visible wave speed is.
(TIF)

**S19 Fig. A flash wave as recorded from different angles surrounding a cylindrical school around its length axis.** The wave pattern is visible from every angle. In particular, when looking from below (angles of 240, 270 and 300 degrees), an anti-flash wave can be seen.
(TIF)

**S20 Fig. A flash wave as recorded from different angles horizontally around a ball-shaped school.** The wave pattern is clearly seen and gets distorted the more we apply perpendicular angles to the wave distance (plots 90 and 270). Even in the perpendicular angles, we can see a dynamic of monotone growth/decay of the flashes.
(TIF)

**S21 Fig. A flash wave as recorded from different angles surrounding a ball shaped school around its length axis.** The wave pattern is visible from every angle. In particular, when looking from below (angles of 240, 270 and 300 degrees), an anti-flash wave can be seen.
(TIF)

**S22 Fig. The four measurements for a shoaling scenario.** The fish density and the observed flash signals are clear but less significant than in the aligned schools. The flash-change signal cannot be seen.
(TIF)

**S23 Fig. Another example of shoaling.** Compared to S22 Fig, the patterns of the noises slightly differ but the signals are the same.
(TIF)

**S24 Fig. Noise sensitivity.** a. The number of Boids reflecting light towards an observer which moves horizontally crossing the center of the mass of an oval group of 1000 "relaxed" Boids under light distribution resembles an ideal Snel's window. b. is a table of the average number of visible flashes towards a horizontal observer perpendicular to the school's swimming direction. The columns represent different noises in the roll of the fish and the rows show different noises in the direction of the velocity of the fish. The roll noises have stronger effects on the flashes. Flashes in the high velocity noise are also due to strong torsion (changes in roll) required to "correct" the noise. A similar analysis with roll always set to zero gave zero flashes in all configurations. c. shows the same scenario of plot b, except that the observer is placed facing the school's direction. The trend of dependency on the roll-noise is preserved. The combined effect of both noises has a stronger relative influence on the number of flashes and the total numbers of flashes are lower in comparison to each one of them separately.
(TIF)

**S25 Fig. Sensitivity to rules weights.** In our model, an agent in emergency mode is prioritizing cohesion and alignment over repulsion for the sake of avoiding separation from the group in strong attacks. In our tests, which were consisted of short local attacks, there is no difference in the speed of the flash wave between copy response with weights changes (left column) and copy response without weights changes (right column).
(TIF)

**S1 Video. Shimmering waves.**
(AVI)

**S2 Video. Attack-like camera angle.**
(MP4)

**S1 File. Consists of the supplementary information text.**
(PDF)

## Author Contributions

**Conceptualization:** Assaf Pertzelan, Gil Ariel, Moshe Kiflawi.

**Formal analysis:** Assaf Pertzelan, Gil Ariel, Moshe Kiflawi.

**Investigation:** Gil Ariel, Moshe Kiflawi.

**Methodology:** Assaf Pertzelan, Gil Ariel, Moshe Kiflawi.

**Resources:** Gil Ariel, Moshe Kiflawi.

**Software:** Assaf Pertzelan, Gil Ariel.

**Supervision:** Gil Ariel, Moshe Kiflawi.

**Validation:** Assaf Pertzelan, Gil Ariel, Moshe Kiflawi.

**Visualization:** Assaf Pertzelan.

**Writing – original draft:** Assaf Pertzelan, Gil Ariel, Moshe Kiflawi.

**Writing – review & editing:** Assaf Pertzelan, Gil Ariel, Moshe Kiflawi.

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
