## [Decision Letter · Decision Letter 0]

20 Feb 2023

PONE-D-23-00798SCHOOLING OF LIGHT REFLECTING FISHPLOS ONE

Dear Dr. Pertzelan,

Thank you for submitting your manuscript to PLOS ONE. After careful consideration, we feel that it has merit but does not fully meet PLOS ONE’s publication criteria as it currently stands. Therefore, we invite you to submit a revised version of the manuscript that addresses the points raised during the review process.

We look forward to receiving your revised manuscript.

Kind regards,

Pilwon Kim

Academic Editor

PLOS ONE

"NO authors have competing interests."

Reviewers' comments:

Reviewer's Responses to Questions

**Comments to the Author**

1. Is the manuscript technically sound, and do the data support the conclusions?

Reviewer #1: Yes

Reviewer #2: Yes

2. Has the statistical analysis been performed appropriately and rigorously? 

Reviewer #1: N/A

Reviewer #2: Yes

3. Have the authors made all data underlying the findings in their manuscript fully available?

Reviewer #1: Yes

Reviewer #2: Yes

4. Is the manuscript presented in an intelligible fashion and written in standard English?

Reviewer #1: Yes

Reviewer #2: Yes

5. Review Comments to the Author

Reviewer #1: I find this work very interesting and with an important contribution to the field. The methods are sound and the text well written. I have some concerns about some parts and suggestions that in my opinion may help the paper’s impact. Please find below my comments.

1) I am not convinced that the view in Figure 8 provides support to the fact that visual cues of shimmering waves are or aren’t sensed by the fish. The group density has a large effect on such view in a real school, especially when individuals are not flat but the body of ones neighbours covers a large part of its field of view. Also, the emergent group density during schooling is the simulations is not mentioned. At the same time, the anticipation mode assumed in line 445 is supported by empirical findings that show that alarmed fish do reduce their response-latency when there are sensing predator risk at the environment.:

Herbert-Read, J.E., Rosén, E., Szorkovszky, A., Ioannou, C.C., Rogell, B., Perna, A., Ramnarine, I.W., Kotrschal, A., Kolm, N., Krause, J. and Sumpter, D.J., 2017. How predation shapes the social interaction rules of shoaling fish. Proceedings of the Royal Society B: Biological Sciences, 284(1861), p.20171126.

Thus, early signals of predation can also be chemical cues. In terms of the results of section 4, the earlier signal results in the wave accelerating, which is quite intuitive. I thus think that section 4 should be revised to make its take away message clearer or otherwise incorporated as a special case of section 2 and have the machine learning methodology details of section 3 moved earlier before section 2 and the presentation of the sections (line 227).

2) Lines 379-392 are a bit too technical for non-experts on machine learning. I think the authors should either give some more definitions (dropout, patience, introduction of the layers, etc) or move some of the details in the supplementary.

3) The discussion is a bit poor. The only paragraph that relates to literature is the one on section 4, that I think is the less strong part of the paper’s contribution. Extending the discussion I think can highly benefit the paper’s impact.

4) Lines 123-124: there has been previous work on agitation waves of starlings where measurements of luminance were used to measure wave intensity and propagation. I think this work is highly relevant to the methodology presented here and should be included:

Hemelrijk, C. K., Costanzo, A., Hildenbrandt, H., & Carere, C. (2019). Damping of waves of agitation in starling flocks. Behavioral Ecology and Sociobiology, 73, 1-7.

5) Lines 172-173: I think that the decision to give priority to cohesion and alignments over repulsion in evasive steps needs a reference or an explanation.

6) Line 309: it would be nice to have the time also in seconds here rather than only steps.

7) Lines 338-340: it is a bit unclear from the text what differentiation the authors mean. It is clear after reading the tasks, so I would suggest revising these sentences to clarify it.

8) Why does the group size decrease from 15000 to 1000 agents from 2 to 3 and 4?

9) Line 345, 360, 362 etc: I find the ‘class’ terminology confusing and unnecessary. Just referring to scenarios throughout may be more intuitive.

Minor comments:

- Line 131: two opening quotes in ‘schooling’ state

- Figure 4 and 9: the colorscale labels are missing

- Line 369: closing parenthesis missing

- Line 484: figure ref mistake

- Lines 228, 231, 232: I guess it means sections 2, 3, and 4 respectively

- Lines 524: Closing parenthesis missing

Reviewer #2: This paper presents an agent-based model of collective animal behavior designed to replicate the shimmering wave of pelagic fish schools, which propagate flashes of sunlight reflected off fish skin. As an interested reader, I found the methodology of flash observations from the external observer to be innovative. Also, the hypothesis that "the heightened state of anticipation is caused by changes in the pattern of reflected light, as perceived by school members found downstream of the propagating wave" is intriguing. However, I have some comments and suggestions regarding the interpretation of the results:

(1)This paper was conducted entirely through computer simulations, without using empirical or experimental data on real shimmering waves. Therefore, there are likely to be several limitations that should be addressed in future studies with real animal data, but the authors provide almost no descriptions regarding the lack of such data.

(a)For example, the authors suggest that fish are able to perceive shimmering waves, which enables them to rapidly propagate waves and detect the presence of predators. While this is an interesting result from computer simulation, it is not experimentally substantiated. Emphasizing that this can be tested with real animal groups would make the paper more persuasive. Indeed, even with real data, if available, the speeding up of wave propagation caused by flash perception by fish could be observed as a change in the slope of fish density and observed flashes in the time-distance plot (Fig. 9).

(b)One weak point of this study is that the proposed model includes too many parameters, which can make the model behavior difficult to interpret and understand. It is unclear whether, for example, different weights of the three rules (alignment, repulsion, and cohesion) are necessary for different group states (schooling, evasive, predator response, and copy response). While the authors investigated model behavior by varying some parameters such as latency, they fixed other parameters. To determine whether these parameters are essential, comparisons with real fish school data would be necessary.

(2) I am interested in the statement regarding criticality in the Discussion (lines 524-527), but I believe the authors should make less strong assessments. Empirical observations of criticality in collective animal behavior have been accumulated (e.g., Cavagna et al., PNAS, 2010), and even clear power-law behavior was found recently (Gómez-Nava et al., Nat. Phys., 2023), although it was not about wave propagation. I therefore would say that non-local signaling by flashes alleviates the need to assume that the school is poised near criticality, “at least” when it comes to wave propagation involving drastic high-speed turns with almost 180 degree rotation.

In relation to this, I wonder whether there could be other advantages of non-local signaling when compared to criticality, such as avoiding false alarms. A system at a critical point is known to be highly sensitive to external perturbations, and this sensitivity probably increases the chances of 'false alarms,' which can be a disadvantage for the system when induced collective behavior requires large energetic costs to perform, such as the drastic turns observed in the shimmering wave. However, when individuals perceive flashes to perform turns, they should perceive it as a mass of groupmates (a conspicuous area of flashes of light) not as individual group mates, which may enable them to avoid reacting to the decisions of a single member, decreasing the chance of false alarms.

Minors:

Fig. 1: Where and how did you obtain this video? Was it recorded by yourselves? Have you published any papers on this field observation elsewhere? If so, please refer to them. Additionally, would it be possible to include this video as supplemental material to help readers better understand what a shimmering wave is?

Line 140: Would it be more accurate to say "corresponding to the direction of the fish's back" instead of "corresponding to the back of the fish"?

Line 170: Is there any empirical evidence supporting the maximum speed mentioned?

Lines 199-213: It would be beneficial to move Fig. S3 to the main text to assist readers in understanding the calculation of light reflected towards the external observer.

References

Cavagna, A., Cimarelli, A., Giardina, I., Parisi, G., Santagati, R., Stefanini, F., & Viale, M. (2010). Scale-free correlations in starling flocks. Proceedings of the National Academy of Sciences, 107(26), 11865-11870. doi: doi.org/10.1073/pnas.1005766107

Gómez-Nava, L., Lange, R. T., Klamser, P. P., Lukas, J., Arias-Rodriguez, L., Bierbach, D., ... & Romanczuk, P. (2023). Fish shoals resemble a stochastic excitable system driven by environmental perturbations. Nature Physics, 1-7. doi: doi.org/10.1038/s41567-022-01916-1

6. PLOS authors have the option to publish the peer review history of their article (what does this mean?). If published, this will include your full peer review and any attached files.

Reviewer #1: No

Reviewer #2: No

---

## [Author Response · Author response to Decision Letter 0]

24 May 2023

Dear Editor,

We thank the referees for their positive reviews and constructive comments. All comments have been addressed. Following the suggestion of reviewer 1, the layout of the paper was changed. Specifically, we first describe generation of flash waves in section 3 (no flash response) and 4 (with flash response). Then, our demonstration that using machine learning, school traits can be inferred from flash patterns is presented in n section 5. Also, as requested, we are hereby stating that the authors have declared that no competing interests exist.

Below we address the specific comments raised by the referees. 

Sincerely,

Assaf Pertzelan, Gil Ariel and Moshe Kiflawi

Editorial comments:

Additional requirements

Reply: done. For convenience, the formatting changes were done without ‘track changes’.

"NO authors have competing interests."

Reply: done

Reply: done

  

Reviewer #1 

I find this work very interesting and with an important contribution to the field. The methods are sound and the text well written. I have some concerns about some parts and suggestions that in my opinion may help the paper’s impact. Please find below my comments.

1) I am not convinced that the view in Figure 8 provides support to the fact that visual cues of shimmering waves are or aren’t sensed by the fish. The group density has a large effect on such view in a real school, especially when individuals are not flat but the body of ones neighbours covers a large part of its field of view. 

Reply: The role of figure 8 (now 7) is only to provide indication that flashes may potentially be observed from within a school – a possibility that was not discussed earlier in the text. Indeed, the view in real schools may be different. For this reason, our results are only a proof of concept for such a putative mechanism that deserves further study. We have changed the figure caption to clarify this point and noted that the view from real schools may be different due to several effects, including occlusions. 

1b) Also, the emergent group density during schooling is the simulations is not mentioned. 

Reply: The inter-distance between agents (40 cm, 4BL, corresponding to ~16 fish per m3) is specified in the methods section (line 251). For demonstration purposes we chose a relatively sparse group. However, Fig 1 (bottom-left) provides a qualitative indication that even in denser groups, flashes can transverse a relatively long distance before they are occluded. The effect of density on occlusions is also discussed in the SI.

1c) At the same time, the anticipation mode assumed in line 445 is supported by empirical findings that show that alarmed fish do reduce their response-latency when there are sensing predator risk at the environment:

Herbert-Read, J.E., Rosén, E., Szorkovszky, A., Ioannou, C.C., Rogell, B., Perna, A., Ramnarine, I.W., Kotrschal, A., Kolm, N., Krause, J. and Sumpter, D.J., 2017. How predation shapes the social interaction rules of shoaling fish. Proceedings of the Royal Society B: Biological Sciences, 284(1861), p.20171126.

Reply: We added the reference. Thank you.

1d) Thus, early signals of predation can also be chemical cues. In terms of the results of section 4, the earlier signal results in the wave accelerating, which is quite intuitive. I thus think that section 4 should be revised to make its take away message clearer or otherwise incorporated as a special case of section 2 and have the machine learning methodology details of section 3 moved earlier before section 2 and the presentation of the sections (line 227).

Reply: We thank the reviewer for this important comment. The general layout of the paper was changed accordingly.

2) Lines 379-392 are a bit too technical for non-experts on machine learning. I think the authors should either give some more definitions (dropout, patience, introduction of the layers, etc.) or move some of the details in the supplementary.

Reply: In order to make the section clearer for non-experts, we moved the description of technical aspects to the supplementary (‘classification model’ section) and added reference to the text. Lines 377-404 were moved to the same section as well (old numbering. Sub sections ‘our model’, ‘evaluation of our model’, and ‘random noise’). Instead, we provide a basic description free of ML technical jargon.

3) The discussion is a bit poor. The only paragraph that relates to literature is the one on section 4, that I think is the less strong part of the paper’s contribution. Extending the discussion I think can highly benefit the paper’s impact.

Reply: We extended the discussion.

4) Lines 123-124: there has been previous work on agitation waves of starlings where measurements of luminance were used to measure wave intensity and propagation. I think this work is highly relevant to the methodology presented here and should be included:

Hemelrijk, C. K., Costanzo, A., Hildenbrandt, H., & Carere, C. (2019). Damping of waves of agitation in starling flocks. Behavioral Ecology and Sociobiology, 73, 1-7.

Reply: Thank you. We added a note related to this study to the introduction (line 51).

5) Lines 172-173: I think that the decision to give priority to cohesion and alignments over repulsion in evasive steps needs a reference or an explanation.

Reply: Please note that during evasive steps cohesion and alignment do not get full priority over repulsion steps, but only the relative weights change. This change prevents sporadic loss of group members. In order to better address this point, we verified that reducing the strength of cohesion does not change the overall dynamics of the shimmering waves. See the new fig S3.2 in the SI.

6) Line 309: it would be nice to have the time also in seconds here rather than only steps.

Reply: Changed. We added “corresponding to 3.22 seconds (see Table 1)”. 

7) Lines 338-340: it is a bit unclear from the text what differentiation the authors mean. It is clear after reading the tasks, so I would suggest revising these sentences to clarify it.

Reply: Changed to “Here, we take a complementary approach and demonstrate that, in of themselves, flash patterns contain enough information to enable an observer to differentiate between behavioral scenarios; i.e., the observer could infer what the fish are doing, without seeing the actual fish”.

8) Why does the group size decrease from 15000 to 1000 agents from 2 to 3 and 4?

Reply: Our motivation in section 3 is to establish the existence of emergent behavior on a large scale. Having demonstrated that, in section 5 (previously 4), we sought to demonstrate the ability to extract information from this emergent behavior. The smaller N used in section 5 is due to technical reasons and reduction of the computation cost involved in producing large datasets of synthetic data. As we are assuming fairly simple internal dynamics on the local scale (e.g. the wave is either propagating or not), our underlying assumption is that emergent patterns that appear with small N would also appear with larger N. This point is explained in the revised text.

9) Line 345, 360, 362 etc: I find the ‘class’ terminology confusing and unnecessary. Just referring to scenarios throughout may be more intuitive.

Reply: We believe it would be better to stay with the term ‘class’, which is the standard terminology in the context of classification method. We use the term ‘scenario’ to describe a ’what we told the simulation to do’ (e.g. direct-and-copy) and ‘class’ as the common term in the context of our technical method. We slightly revised the text in order to better explain this point.

Minor comments:

- Line 131: two opening quotes in ‘schooling’ state

- Figure 4 and 9: the colorscale labels are missing

- Line 369: closing parenthesis missing

- Line 484: figure ref mistake

- Lines 228, 231, 232: I guess it means sections 2, 3, and 4 respectively

- Lines 524: Closing parenthesis missing

Reply: All corrected.

 

Reviewer #2

This paper presents an agent-based model of collective animal behavior designed to replicate the shimmering wave of pelagic fish schools, which propagate flashes of sunlight reflected off fish skin. As an interested reader, I found the methodology of flash observations from the external observer to be innovative. Also, the hypothesis that "the heightened state of anticipation is caused by changes in the pattern of reflected light, as perceived by school members found downstream of the propagating wave" is intriguing. However, I have some comments and suggestions regarding the interpretation of the results:

(1) This paper was conducted entirely through computer simulations, without using empirical or experimental data on real shimmering waves. Therefore, there are likely to be several limitations that should be addressed in future studies with real animal data, but the authors provide almost no descriptions regarding the lack of such data.

(a) For example, the authors suggest that fish are able to perceive shimmering waves, which enables them to rapidly propagate waves and detect the presence of predators. While this is an interesting result from computer simulation, it is not experimentally substantiated. Emphasizing that this can be tested with real animal groups would make the paper more persuasive. Indeed, even with real data, if available, the speeding up of wave propagation caused by flash perception by fish could be observed as a change in the slope of fish density and observed flashes in the time-distance plot (Fig. 9).

Reply: We agree that empirical study is a necessary next-step of our simulation-based study. We added some comments to this effect in the discussion.

(b) One weak point of this study is that the proposed model includes too many parameters, which can make the model behavior difficult to interpret and understand. It is unclear whether, for example, different weights of the three rules (alignment, repulsion, and cohesion) are necessary for different group states (schooling, evasive, predator response, and copy response). While the authors investigated model behavior by varying some parameters such as latency, they fixed other parameters. To determine whether these parameters are essential, comparisons with real fish school data would be necessary.

Reply: Indeed, the model has many parameters. Unfortunately, we could not come up with a simpler model that successfully reproduces the range of behaviors we are interested in. Some parameter values were obtained (at least approximately) from the literature, as detailed in Table 1. Other, were found by trial and error. In the SI, we describe some sensitivity analysis (e.g. rule weights), showing that the main conclusions do not change qualitative in a range of examined parameters. In order to better address this point, we also added (SI fig S3.2) a comparison between emergency response with and without prioritizing the cohesion and alignment over repulsion). Additionally, we study the effect of different latencies in the SI and figs S1, S13, S14.

(2) I am interested in the statement regarding criticality in the Discussion (lines 524-527), but I believe the authors should make less strong assessments. Empirical observations of criticality in collective animal behavior have been accumulated (e.g., Cavagna et al., PNAS, 2010), and even clear power-law behavior was found recently (Gómez-Nava et al., Nat. Phys., 2023), although it was not about wave propagation. I therefore would say that non-local signaling by flashes alleviates the need to assume that the school is poised near criticality, “at least” when it comes to wave propagation involving drastic high-speed turns with almost 180 degree rotation.

In relation to this, I wonder whether there could be other advantages of non-local signaling when compared to criticality, such as avoiding false alarms. A system at a critical point is known to be highly sensitive to external perturbations, and this sensitivity probably increases the chances of 'false alarms,' which can be a disadvantage for the system when induced collective behavior requires large energetic costs to perform, such as the drastic turns observed in the shimmering wave. However, when individuals perceive flashes to perform turns, they should perceive it as a mass of groupmates (a conspicuous area of flashes of light) not as individual group mates, which may enable them to avoid reacting to the decisions of a single member, decreasing the chance of false alarms. 

Reply: We thank the referee for this important point. Accordingly, the discussion regarding criticality was extended in the discussion section. In addition, the following reference, which are related to this discussion, were added:

Bialek W, Cavagna A, Giardina I, Mora T, Pohl O, Silvestri E, et al. Social interactions dominate speed control in poising natural flocks near criticality. PNAS. 2014;111: 7212–7217. doi:10.1073/pnas.1324045111

Romanczuk P, Daniels BC. Phase Transitions and Criticality in the Collective Behavior of Animals ? Self-Organization and Biological Function. Order, Disorder and Criticality. WORLD SCIENTIFIC; 2022. pp. 179–208. doi:10.1142/9789811260438_0004

Cavagna A, Cimarelli A, Giardina I, Parisi G, Santagati R, Stefanini F, et al. Scale-free correlations in starling flocks. Proceedings of the National Academy of Sciences. 2010;107: 11865–11870. doi:10.1073/pnas.1005766107

Gómez-Nava L, Lange RT, Klamser PP, Lukas J, Arias-Rodriguez L, Bierbach D, et al. Fish shoals resemble a stochastic excitable system driven by environmental perturbations. Nat Phys. 2023; 1–7. doi:10.1038/s41567-022-01916-1

Minors:

Fig. 1: Where and how did you obtain this video? Was it recorded by yourselves? Have you published any papers on this field observation elsewhere? If so, please refer to them. Additionally, would it be possible to include this video as supplemental material to help readers better understand what a shimmering wave is?

Reply: The videos were recorded by us but not published. We added the details to the figure caption and the videos to the supplementary information. 

Line 140: Would it be more accurate to say "corresponding to the direction of the fish's back" instead of "corresponding to the back of the fish"?

Reply: Changed.

Line 170: Is there any empirical evidence supporting the maximum speed mentioned?

Reply: The line number may be wrong (there is no reference to maximum speed there). The maximum speed is within the range reported by Domenici and Hale (2019). We added a clarification in Table 1. 

Lines 199-213: It would be beneficial to move Fig. S3 to the main text to assist readers in understanding the calculation of light reflected towards the external observer.

Reply: Done

---

## [Decision Letter · Decision Letter 1]

26 Jun 2023

PONE-D-23-00798R1SCHOOLING OF LIGHT REFLECTING FISHPLOS ONE

Dear Dr. Pertzelan,

Thank you for submitting your manuscript to PLOS ONE. After careful consideration, we feel that it has merit but does not fully meet PLOS ONE’s publication criteria as it currently stands. Therefore, we invite you to submit a revised version of the manuscript that addresses the points raised during the review process.

We look forward to receiving your revised manuscript.

Kind regards,

Pilwon Kim

Academic Editor

PLOS ONE

Journal Requirements:

Reviewers' comments:

Reviewer's Responses to Questions

**Comments to the Author**

1. If the authors have adequately addressed your comments raised in a previous round of review and you feel that this manuscript is now acceptable for publication, you may indicate that here to bypass the “Comments to the Author” section, enter your conflict of interest statement in the “Confidential to Editor” section, and submit your "Accept" recommendation.

Reviewer #1: All comments have been addressed

Reviewer #2: All comments have been addressed

2. Is the manuscript technically sound, and do the data support the conclusions?

Reviewer #1: Yes

Reviewer #2: Yes

3. Has the statistical analysis been performed appropriately and rigorously? 

Reviewer #1: N/A

Reviewer #2: Yes

4. Have the authors made all data underlying the findings in their manuscript fully available?

Reviewer #1: Yes

Reviewer #2: Yes

5. Is the manuscript presented in an intelligible fashion and written in standard English?

Reviewer #1: Yes

Reviewer #2: Yes

6. Review Comments to the Author

Reviewer #1: The authors did an excellent job revising the manuscript according to the given comments, I find the paper very improved and I am happy to recommend its acceptance after a few minor comments being considered, please find them below:

- Figure 3 and Lines 262-271: I find the caption and figure a bit difficult to understand, I think they could be improved a bit: (a) which is the yellow cone? It is only in 3a, (b) Which direction is towards us? (c) what are the dotted blue lines in b? (d) perhaps an annotation for t and t-1 can be added on the figure, (e) what are the yellow arrows representing?

- Line 413: ‘recreate’

- Line 416: ‘in and of themselves’ is confusing appearing before the subject ‘flash patterns’

- Lines 419-420: I think the sentence needs revision, is ‘rather than’ what the authors mean?

- Lines 422: ‘of’ the individual

- Line 424: ‘which is and observer’ ?

- Line 426: ‘that’ before ‘a stationary’ can improve the sentence

- Figure 8 & Lines 452-455: the figure and caption should be reordered to follow the new order of the text (a - attack direction, b – shape, c – motion rules). Also, references to Figure 8b and 8c can be added to the relevant text sessions (line 434 and 439).

- Line 503: ‘attack’

- Perhaps Figures 9-11 could be merged into 1 figure as A, B, C subplots.

- Line 581: I would replace ‘are’ with ‘may’, since its hasn’t been tested in real fish.

Reviewer #2: The authors have done a thorough work of responding to my comments. The previous version was a nice contribution, and the revision have made it more so.

The quality (resolution) of the figures is quite low and some of them are invisible. They should be improved at the proof.

7. PLOS authors have the option to publish the peer review history of their article (what does this mean?). If published, this will include your full peer review and any attached files.

Reviewer #1: No

Reviewer #2: No

---

## [Author Response · Author response to Decision Letter 1]

5 Jul 2023

Dear Editor and Reviewers,

We thank the referees for their positive reviews and constructive comments. All comments have been addressed. We have accepted all the (minor) comments of Reviewer #1 and made changes accordingly. In particular, we redrew figure #3 in order to make it clearer. Regarding the resolution of the figures (cf. Reviewer #2), the resolution of the figures in the auto-generated PDF appeared to be lower compared to the resolution of the figures we originally submitted. Figures with a higher resolution will be generated per request from the production editor.

Once again, we sincerely appreciate the efforts of the reviewers in evaluating our manuscript. Their feedback has been instrumental in refining our work, and we hope that the revised version now meets the standards of PLoS One.

Yours sincerely, 

Assaf Pertzelan, Moshe Kiflawi, and Gil Ariel

---

## [Editor Report · Decision Letter 2]

10 Jul 2023

SCHOOLING OF LIGHT REFLECTING FISH

PONE-D-23-00798R2

Dear Dr. Pertzelan,

We’re pleased to inform you that your manuscript has been judged scientifically suitable for publication and will be formally accepted for publication once it meets all outstanding technical requirements.

Kind regards,

Pilwon Kim

Academic Editor

PLOS ONE

---

## [Editor Report · Acceptance letter]

13 Jul 2023

PONE-D-23-00798R2 

Schooling of light reflecting fish 

Dear Dr. Pertzelan:

I'm pleased to inform you that your manuscript has been deemed suitable for publication in PLOS ONE. Congratulations! Your manuscript is now with our production department. 

Kind regards, 

on behalf of

Professor Pilwon Kim 

Academic Editor

PLOS ONE